# Synthesis and Antiproliferative Activity against Cancer Cells of Indole-Aryl-Amide Derivatives

**DOI:** 10.3390/molecules28010265

**Published:** 2022-12-28

**Authors:** Junwei Zhao, Jacopo Carbone, Giovanna Farruggia, Anna Janecka, Luca Gentilucci, Natalia Calonghi

**Affiliations:** 1Department of Chemistry “G. Ciamician”, University of Bologna, Via Selmi 2, 40126 Bologna, Italy; 2Department of Pharmacy and Biotechnology (FaBiT), University of Bologna, Via Irnerio 48, 40126 Bologna, Italy; 3Department of Biomolecular Chemistry, Medical University of Lodz, Mazowiecka 6/8, 92-215 Lodz, Poland

**Keywords:** indole nucleus, κ-opioid receptor, molecular docking, anticancer, colon cancer

## Abstract

Indoles constitute a large family of heterocyclic compounds widely occurring in nature which are present in a number of bioactive natural and synthetic compounds, including anticancer agents or atypical opioid agonists. As a result, exponential increases in the development of novel methods for the synthesis of indole-containing compounds have been reported in the literature. A series of indole-aryl amide derivatives **1**–**7** containing tryptamine or an indolylacetic acid nucleus were designed, synthesized, and evaluated as opioid ligands. These new indole derivatives showed negligible to very low affinity for μ- and δ-opioid receptor (OR). On the other hand, compounds **2**, **5** and **7** showed Ki values in the low μM range for κ-OR. Since indoles are well known for their anticancer potential, their effect against a panel of tumor cell lines was tested. The target compounds were evaluated for their in vitro cytotoxicity in HT29, HeLa, IGROV-1, MCF7, PC-3, and Jurkat J6 cells. Some of the synthesized compounds showed good activity against the selected tumor cell lines, with the exception of IGROV1. In particular, compound **5** showed a noteworthy selectivity towards HT29 cells, a malignant colonic cell line, without affecting healthy human intestinal cells. Further studies revealed that **5** caused the cell cycle arrest in the G1 phase and promoted apoptosis in HT29 cells.

## 1. Introduction

The indole nucleus is an important element of many natural and synthetic molecules with significant biological activities, with potential applications against cancer, HIV, inflammation, tuberculosis, hypertension, diabetes, and against viral, fungal, and microbial infections [1,2,3,4]. In addition, the indole moiety is included within the structure of atypical opioid receptor (OR) agonists. The indole alkaloid mitragyna (Figure 1), isolated from the tree of the Rubiaceae family *Mitragyna speciosa*, and pericine (Figure 1), found in the tree *Picralima nitida,* were shown to act as a partial μ-OR agonists [5].

In 2002, Saito et al. isolated the tetrapeptide CJ-15,208, c[Phe-D-Pro-Phe-Trp] from the fungus *Ctenomyces serratus* [6], which revealed itself to be a modestly selective κ/μ opioid ligand. Subsequently, other structurally correlated Trp-containing linear [7] or cyclic peptides [8] came to the spotlight for their unusual agonist or antagonist opioid activity. Discovered independently from CJ-15,208, the cyclopentapeptide c[Phe-Gly-Tyr-D-Pro-D-Trp] showed itself to be selective mu agonist with nM activity [9]. This cyclopeptide displayed sufficient metabolic stability to be effective after peripheral administration and demonstrated a therapeutic potential as a novel analgesic agent in controlling visceral pain [10]. Very recently, the cyclopeptide c[Phe-Gly-β-Ala-D-Trp] (LOR17) was found to be a κ-OR agonist with functional selectivity toward G protein signaling and eliciting antinociceptive and antihypersensitivity effects in different animal models [11].

For the absence of a cationizable amino group (Figure 1), generally regarded as the fundamental pharmacophore [12], the structure of the Trp-containing peptides CJ-15,208, c[FGYpw], and LOR17, described above, appeared to differ significantly from the classic opioid ligands [13]. Structure–activity relationship [11,14] and docking analyses [15,16] strongly supported that the bioactivity mainly resides in the aromatic side chains of the Trp-Phe dipeptide, i.e., the indole and phenyl rings.

In this perspective, we became interested in the synthesis of minimalist compounds containing indole and phenyl rings in diverse structural combinations. For this purpose, we designed the amides **1**–**7** based either on a tryptamine or an indolylacetic acid nucleus, that we analyzed as potential opioid ligands. As it turned out, the compounds showed modest affinities for the ORs. However, some peptides demonstrated a certain selectivity for κ-OR.

On the other hand, indole derivatives seem to be important in cancer via acting through various mechanisms [1,2,17]. Examples of anticancer drugs are the indole alkaloids vincristine and vinblastine (Figure 1), obtained from *Catharanthus roseus*.

Vinblastine is applied to the treatment of various kinds of cancer, such as Kaposi’s sarcoma, Hodgkin’s disease, non-Hodgkin’s lymphoma, and testicular or breast cancer [18,19]. Besides, the indole unit was found to be a privileged scaffold for drug discovery [19]. Hence, the compounds **1**–**7** were investigated for their potential cytotoxic activity against a panel of cancer cell lines.

## 2. Results and Discussion

### 2.1. Chemistry

The indole derivatives were obtained by an expedient two-step procedure (Figure 1).

The structures of substituents R and R^1^ are shown in Figure 2.

Reductive amination of tryptamine [20] followed by coupling with 3-(4-hydroxyphenyl)propanoic acid or 2-(4-aminophenyl)acetic acid, under standard conditions gave the structures **1** and **2**, respectively; compound **3** was obtained when a two-fold excess of 2-(4-aminophenyl)acetic acid was used (route A).

The reductive amination of 4-(aminomethyl)aniline with either 4-hydroxybenzaldehyde, 2-oxoacetamide, or 3-hydroxypropanal, followed by amide bond formation with 2-(1H-indol-3-yl)acetic acid, gave **5**–**7**. Compound **4** was obtained by direct reaction between 4-(aminomethyl)aniline and 2-(1H-indol-3-yl)acetic acid (route B). The compounds were isolated in good yield (78–85% after two steps) by flash chromatography over silica gel and analyzed by mass and NMR spectroscopy. Copies of ^1^H NMR of **1**–**7** in CDCl_3_, 400 MHz, are reported in the Appendix A).

### 2.2. Determination of the Affinity for Opioid Receptors

In order to evaluate the affinity of the compounds for μ-, δ-, κ-ORs, radioligand binding assays were performed. Displacement of radioligands, [^3^H]DAMGO (μ-OR selective), [^3^H]deltorphin-2 (δ-OR selective), and [^3^H]U-69593 (κ-OR selective) by the synthetic analogs was investigated in the commercial membranes of CHO cells transfected with human recombinant ORs. The results are summarized in Table 1. All compounds showed negligible to very modest affinity for μ-ORs and δ-ORs; on the other hand, **2**, **5**, and **7**, showed K_i_ values in the low μM range for κ-ORs. Due to the modest receptor affinities, the functional activity of the compounds was not investigated.

In spite of the moderate affinity for the OR, the clear selectivity for κ-OR shown by some compounds prompted us to investigate the plausible ligand–receptor interactions by molecular docking simulations. Indeed, these structure lack in the classic, principal pharmacophore of opioid ligands, i.e., the protonable amine.

### 2.3. Molecular Docking of 5 and 2 in κ-OR

Docking simulations at κ-OR were performed for **2** and **5**, characterized by the comparatively lower Ki values, utilized as prototypic compounds of the derivatives of tryptamine and of indolyl acetic acid, respectively (Table 1). Simulations were performed with AutoDock Vina (see the Experimental Section), using the structure of the human κ-OR in complex with the tetrahydro-isoquinolinecarboxamide ligand JDTic, PDB ID: 4DJH [21]. We preferred this structure respect to crystal structure of human κ-OR in complex with the epoxymorphinan opioid agonist MP1104, PDB ID: 6B73, for the higher degree of similarity between the polycyclic and flexible structures of JDTic, **5**, and **2**. Besides, the structure 4DJH was utilized for performing a molecular docking of the salvinorin A analogue RB-64. Salvinorin, **2**, and **5**, share a relevant feature as compared to other OR ligands in that they lack the tyramine primary pharmacophore [12,22]. The ligand–receptor complexes were obtained by a systematic conformer search, followed by geometry optimization in explicit TIP3P water molecules.

For each ligand, independent docking runs were performed, and the results were scored and clustered. Protein–ligand complexes from selected docked poses were minimized in explicit TIP3P water molecules and equilibrated by molecular dynamics. More details can be found in the literature [23] and in the experimental section.

The side views of the best-scoring poses are shown in Figure 3; for alterative top views of the predicted binding poses see also Appendix A.

Apparently, the two compounds adopt different orientations within the receptor. In both complexes, the message-binding region of the receptors, delimited by residues of the transmembrane helices (TM)-2, 3, and 7, appears to be occupied by the indole group. Indeed, in **2** the indole interacts with Asp138, Tyr320, and Val134, while in **5** it interacts with Asp138, Tyr320, Ile316, Ile290, Trp287. The rest of the structures of **5** and **2** adopt alternative orientation within the receptor (Figure 3), occupying “address-recognition” regions that are not conserved across the other ORs, plausibly explaining the preference for κ-OR.

The key role of the indole ring in the formation of the ligand-receptor complex is not completely unexpected. Previously, tryptophan-containing peptides were shown to bind to the opioid receptors albeit lacking a protonable amino group, and the computations suggested that the indole of Trp interacted with Asp138 and other residues of the message-binding region [8,13]. This peculiarity makes the indole-containing compounds a distinct class of unusual opioid ligands. In the crystal structure of κ-OR/JDTic, the protonated amines in both piperidine and isoquinoline moieties of the ligand form salt bridges to the Asp 138 (3.32). In contrast, the molecular docking simulation of the salvinorin derivative RB-64 showed interaction and/or close association with human κ-OR residues Val108, Gln115, Val118, Tyr119, Tyr313, and Tyr320, but not Asp138.

The alternative structures shown in Figure 3 for **2** and **5** may be of help for understanding the difference of the affinity of each compound towards κ-OR. For the indoles of the tryptamine series, it appears that the aniline nitrogen plays a relevant role in stabilizing the complex. The phenol group of **1** and the aniline of **3** are connected to the rest of the structure by means of linkers of increasing length, therefore they seem unable to interact with Tyr139 in a similar way as **2**. As for the derivatives of the of the indolyl acetic acid series, the hydroxy group of phenol in **5** seems important for the hydrogen bond with Lys227C=O, but the aromatic ring is not. Indeed, derivative **7**, having the N-(3-hydroxypropyl) substituent, also shows a similar receptor affinity.

### 2.4. Biological Effects

#### 2.4.1. Effect of the Amides on Cell Viability

To investigate whether the compounds have an antitumor activity, we performed in vitro viability assays against a panel of cancer cell lines: HT29 (human colorectal adenocarcinoma), HeLa (human cervical carcinoma), IGROV-1 (human ovarian carcinoma), MCF7 (human breast cancer), PC-3 (human prostate adenocarcinoma), and Jurkat J6 (human myeloid leukemia). The I407 (human embryonic intestinal) and 3T3 (mouse embryonic fibroblast) cells were used as controls. The growth inhibitory concentration was determined by incubating the cells with increasing concentrations (0.01–500 μM) of the compounds for 24 h. Data obtained from cell growth assays were elaborated to assess the concentration of cell viability (IC_50_) and selectivity indices (SI), and the results are shown in Table 2 and Table 3, respectively.

In general, none of the compounds showed biological activity in IGROV-1. On the other hand, some of the synthesized compounds showed a significant activity towards other cell lines; furthermore, we observed striking differences in the models’ response.

Compound **1**, which is essentially a *N*-benzyltryptamine *N*-capped with a (4-hydroxyphenyl)propanoyl group, was significantly active against two out of six cancer models: HT29, and HeLa, whit IC_50_ values (μM) of 0.31 and 25, respectively. This compound was able to reduce cell growth also in control I407 cells with IC_50_ of 63 μM, and in the 3T3 cell line, but only at a very high dose (IC_50_ = 164.3 µM).

Compound **2** differs from **1** by the substitution of the N-cap with a (4-aminophenyl) acetyl group. This derivative was active against the cancer cell lines MCF7 and PC3, with an IC_50_ (μM) of 0.81 and of 2.13, respectively. However, this compound showed a potent activity when tested in normal cells, since it reduced cell growth in embryonic intestinal I407 cells, IC_50_ = 0.35 µM.

In **3**, the N-cap is represented by two consecutive (4-aminophenyl)acetyl groups. This compound showed a certain activity only in HeLa cancer cells, with an IC_50_ of 5.64 μM. However, **3** was more potent in reducing growth in the healthy model cell 3T3 (IC_50_ = 0.50 µM) and showed significant activity also in I407 (IC_50_ = 75.0 µM).

Compound **4** is a simple amide formed from 4-(aminomethyl)aniline and indolylacetic acid moieties. Nevertheless **4** was active against 3 out of 6 cancer models: HT29, HeLa, and MCF7, whit IC_50_ values (μM) 0.96, 1.87, 0.84, respectively. This compound was also able to reduce cell growth in the embryonic fibroblast cell line 3T3 at low doses (IC_50_ = 0.63 µM).

In **5**, the 4-(aminomethyl)aniline portion carries a further 4-methilphenol group, and the compound was active against 3 cancer cell lines: HT29, PC3 and J6, with IC_50_ values (μM) of 2.61, 0.39, and of 0.37, respectively. This compound was also able to significantly reduce cell growth at in 3T3 cells (IC_50_ = 32 µM).

The activity of **6** in reducing cell growth was limited to PC3 cells, at a comparatively higher dose (IC_50_ = 166 µM), but reduced cell growth in 3T3 cell line (IC_50_ = 32.0 µM). This compound is characterized by the presence of an acetamide group at the 4-(aminomethyl)aniline moiety.

The amide **7** still presents the same scaffold as the amides **4**–**6**, and is equipped with a propan-1-ol group. This indolylacelyl deriuvative was very active in the breast cancer MCF7 (IC_50_ = 0.49 µM) cell line. Interestingly, **7** showed negligible activity against the other cancer cell lines, as well as in heathy cell lines, suggesting a potential utility against breast cancer cells, with little toxicity towards other cell lines, which will be analyzed in due course.

Although not many derivatives were analyzed, a difference in the activity of each compound was found. Plausibly, this may be the result of the chemical features of the amide, aniline, phenol, and aliphatic hydroxy groups, as well as the effect of the presence of electron donation or electron withdrawing groups. However, at the moment, the precise definition of structure-activity trends is beyond our possibility.

Interestingly, the affinity tests for ORs (Table 1) showed that **5** could bind κ-OR with modest affinity but with a good selectivity over μ- and δ-OR; the simulations by molecular docking predicted that **5** binding interactions with the κ-OR receptor involve the indole nucleus. The viability analyses showed that it has good biological activity in human colorectal adenocarcinoma HT29 and no activity in the normal intestinal line, I407. For this reason, we decided to use HT29 as a model to investigate any biological effects of **5**. Indeed, several studies supported that κ-OR plays an important role in cancer development by promoting or inhibiting tumor growth and metastasis and influencing patient prognosis. These observations demonstrated that κ-OR is upregulated in various types of solid tumors, such as liver, colon, non-small cell lung cancer, and other malignancies, and its expression is associated with cancer growth and a poor prognosis [24,25,26].

#### 2.4.2. Compound 5 Inhibits HT29 Colon Cancer Cells Proliferation and Progression in the Cell Cycle

HT29 cells were treated with 1 or 5 μM **5** for 24 h and analyzed by flow cytometry after DNA staining. The results showed that untreated cells progress through the cell cycle. In contrast, **5**-treated HT29 cells accumulated in the G0/G1 phase in a dose-dependent manner, with a nearly complete synchronization (84% ± 0.9) after 5 μM treatment. The flow cytometric analyses, along with the relative phase percentages, are shown in Figure 4.

#### 2.4.3. Effect of 5 on Histone Acetylation

In order to identify acetylated histones, we analyzed the nuclear cell lysates by western blot using a 10% polyacrylammide gel electrophoresis. As shown in Figure 5, the antibody could detect the accumulation of acetylated proteins induced in HT29 cells treated with 5 μM **5** for 6 h. Differences in the density of the bands are thought to reflect differences in protein acetylation levels. Histone acetylation signals were quantified by densitometry and normalized on histone H1 (Figure 5).

HT29 treatment with **5** for 6 h induced different effects on histones acetylation. In particular, **5** induced H4 hyperacetylation, while it did not modify the state of acetylation of H2/H3.

#### 2.4.4. Compound 5 Induces Gene and Protein Expression Modulations

To verify whether the epigenetic modifications were related to changes in gene expression, we proceeded with the qtRT-PCR analysis and western blot of related protein products. Transcription levels of CDKN1B, CDKN1A, and Bax were analysed by RT-PCR performed on cDNA of HT29 control and treated with 5 μM of **5** for 6 h. Genes transcription was normalized to the reference gene *G3PDH* and the relative variations (fold change) are reported in Figure 6A. CDKN1A and Bax genes expression were significantly upregulated by **5** by a factor of 4.16 ± 0.53 (*p* ≤ 0.0001 vs. control), and 2.58 ± 0.11 (*p* ≤ 0.0001 vs. control), respectively; on the other hand, **5** did not modify the gene expression of CDKN1B.

P27, p21, and Bax proteins expressions were analysed by western blot in control HT29 and treated with 5 μM of **5** for 24 h. Compound **5** caused an increase in p21 and Bax, while it did not modify the expression of p27 (Figure 6B).

On resuming the data discussed above, the new indole-aryl amide derivatives, showed null to very modest affinity for μ-OR and δ-OR. On the other hand, **2**, **5**, and **7**, showed K_i_ values in the low μM range for κ-OR. Since indoles are well known for their antitumor potential [1,2], we assayed their effects against a panel of cancer cell lines. In the antiproliferative study, **5** was able to induce selective toxicity toward HT29, a malignant colon cell line, while not affecting healthy human intestine cell.

Possibly, the activity of **5** against HT29 cells might be correlated to its activity towards the κ-ORs. The expression profile of opioid receptors in different cancer cells has also been reported [27], and experimental studies investigating the effects of opioid receptor agonists and antagonists on the proliferation and metastasis of cancer both, in the in vivo and in vitro studies, have received lots of attention.

κ-OR is a member of the G-protein-coupled receptor family, and its natural endogenous ligand is dynorphin, which decreases synaptic transmission by inhibiting adenylate cyclase and voltage-gated calcium channels, activating voltage-gated potassium channels: the results are a decreased neuronal action potential production and a neurotransmitter release [28].

The structure and distribution of κ-OR provide an important clue for its participation in regulating various pathophysiological functions of the body [29]. Current evidence suggests that κ-OR serves a key role in the progression of tumors [30,31]. In addition, κ-OR activates the extracellular signal-regulated kinase (ERK 1/2) and p38 mitogen-activated protein kinase (MAPK) [32,33] and can activate c-Jun amino-terminal kinase (JNK) [34]. Interestingly, prototypical κ-OR antagonists can also activate JNK via a pharmacological process called biased agonism or ligand-directed signaling, whereby ligands can inhibit one intracellular signaling pathway while simultaneously activating another [35,36].

In this preliminary study, we demonstrated that **5** produces an effect on HT29 cells by inhibiting proliferation and promoting apoptosis. The inhibitory effect of **5** on proliferation can be associated with the up-regulation of the anti-proliferative protein p21, while its apoptosis-inductive effect can be associated with the up-regulation of the pro-apoptotic protein Bax. p21 is a cyclin-dependent kinase inhibitor which is mainly associated with the inhibition of CDKs (cyclin-dependent kinases), leading to cell cycle arrest in the context of DNA damage [37,38]. Bax is a member of the Bcl-2 gene family and is a regulator of apoptosis [39]. Bax has been shown to be involved in the p53-mediated pathway of apoptosis, which involves the release of cytochrome c [40]. Through the regulation of apoptosis, Bax is a crucial protein involved in the regulation of cancer cell growth.

## 3. Materials and Methods

### 3.1. General Methods

Standard chemicals were obtained from commercial sources. Purification of the compounds was performed by flash chromatography over silica gel; eluent: cyclohexane/EtOAc 85:15. The purity of the final products was analyzed by reverse-phase (RP) HPLC, performed on Agilent 1100 series apparatus (Agilent, Technologies, Waldbronn, Germany), equipped with a RP column Phenomenex (Torrance, CA, USA) No 00D-4439-Y0 Gemini 3 μm C18 110 Å, LC column 100 mm × 3.0 mm; diode-array detector (DAD) λ = 210 nm and 254 nm; mobile phase: from 9:1 H_2_O/acetonitrile (ACN) containing 0.1% formic acid to 2:8 H_2_O/ACN containing 0.08% formic acid in 20 min, flow rate 1.0 mL min^−1^. ESI-MS was done on a MS single quadrupole HP 1100 MSD detector (Agilent). ^1^H NMR analysis was performed on a Varian Gemini 400 MHz apparatus (Agilent Technologies); peptide samples were dissolved in CDCl_3_ or in DMSO-d6 to the final concentration of 0.01 M and analyzed in 5 mm tubes at rt. Water suppression required the PRESAT presaturation procedure. Chemical shifts (δ) are expressed as p.p.m., using the residual peak of the deuterated solvent as an internal standard (δH DMSO-d6 = 2.50 p.p.m., δH CDCl_3_ = 7.27 p.p.m.).

Opioid radioligands, [^3^H]DAMGO, [^3^H]deltorphin-2 and [^3^H]U-69593, and human recombinant opioid receptors came from PerkinElmer (Krakow, Poland). GF/B glass fiber strips were purchased from Whatman (Brentford, UK).

#### 3.1.1. General Protocol for Reductive Amination

A solution of the amine (1 mmol) and the aldehyde (1 mmol) in MeOH (4 mL) was stirred at rt for 36 h. Solid NaBH_4_ (22.7 mg, 0.6 mmol) was added, and the solution was stirred at rt for 30 min. The solvent was removed at reduced pressure. The reaction mixture was diluted with saturated aqueous NaHCO3 (5 mL), and EtOAc (10 mL). The organic layer was separated and washed sequentially with water (5 mL), brine (5 mL), and dried over Na_2_SO_4_. The solvent was removed at reduced pressure to produce the secondary amine as a brown oil in almost quantitative yield, which was used without further purification. Analytical data for this compound were consistent with those previously reported.

#### 3.1.2. General Protocol for Amide Bond Formation

A mixture of the acid and amino partners (1 mmol each) EDC·HCl (1.1 mmol), and TEA (3 mmol) was stirred in 3:1 DCM/DMF (4 mL) at rt. After 2 h, the solvents were removed at reduced pressure, and the residue was diluted with EtOAc (25 mL). The slurry mixture was washed with 0.1 M HCl (5 mL) and a saturated solution of NaHCO_3_ (5 mL). The organic layer was dried over Na_2_SO_4_ and the solvent was evaporated at reduced pressure. The crude compounds, obtained in almost quantitative yield, were isolated in satisfactory yields by flash chromatography over silica gel (eluent: cyclohexane/EtOAc 85:15).
^1^H NMR (400 MHz, CDCl_3_), (Bn = benzyl, Ind = indole, a = conformer A, b = conformer B) δ 8.07 (bs, 0.53H, IndH1a), 8.04 (bs, 0.47H, IndH1b), 7.55 (d, *J* = 7.6 Hz, 0.47H, IndH4b), 7.45 (d, *J* = 7.6 Hz, 0.53H, IndH4a), 7.38–6.65 (m, 6H + 6H + 1H, ArHa + ArHb + PheOH), 6.92 (bs, 0.53H, IndH2a), 6.85 (bs, 0.47H, IndH2b), 4.64 (s, 1.06H, BnCH_2_a), 4.32 (s, 0.94H, BnCH_2_b), 3.67 (t, *J* = 7.8 Hz, 0.92H, NCH_2_b), 3.47 (t, *J* = 7.1 Hz, 1.08H, NCH_2_a), 3.01 (t, *J* = 7.6 Hz, 0.94H, IndCH_2_b), 2.97–2.85 (m, 0.94H + 1.06H, PheCH_2_b + IndCH_2_a), 2.80 (t, *J* = 7.7 Hz, 1.06H, PheCH_2_a), 2.62 (t, *J* = 7.6 Hz, 0.98H, COCH_2_b), 2.44 (t, *J* = 7.7 Hz, 1.02H, COCH_2_a). MS (ESI) m/z [M + H] ^+^ calcd: 399.2; found: 399.2.^1^H NMR (400 MHz, DMSO-*d*_6_) δ 10.91 (s, 0.6H, IndH1a), 10.79 (s, 0.4H, IndH1b), 7.48 (m, 1H, IndH4a + IndH4b), 7.41–7.35 (m, 1H, IndH7), 7.36–7.28 (m, 2H, BnArH3,5), 7.28–7.16 (m, 3H, BnArH2,4,6), 7.14 (d, *J* = 2.4 Hz, 0.6H, IndH2a), 7.13–7.08 (m, 1H, IndH6), 7.07–7.02 + 6.99–6.91 (m, 2H + 2H, anilineH3,5 + anilineH4,6), 7.07 (d, *J* = 3.3 Hz, 0.4H, IndH2b), 7.02–6.99 (m, 1H, IndH5), 4.62 (s, 2H, BnCH_2_), 3.71 (s, 2H, anilineCH_2_), 3.50 (m, 2H, NCH_2_), 2.78–3.01 (m, 2H, IndCH_2_). MS (ESI) m/z [M + H] ^+^ calcd: 384.2; found: 384.2.^1^H NMR (400 MHz, DMSO-*d*_6_) δ 10.88 (s, 0.62H, IndH1a), 10.78 (s, 0.38H, IndH1b), 10.06 (s, 0.39H, CONHb), 10.03 (s, 0.61H, CONHa), 7.54–7.43 (m, 1H, IndH4a + IndH4b), 7.48–7.44 (m, 1H, IndH7), 7.18–6.86 (m, 3H, ^2^anilineH × 3), 7.38–7.27 (m, 3H, BnH2,4,6), 7.27–7.20 (m, 2H, BnH3,5), 7.18 (d, *J* = 7.5 Hz, 1H, ^2^anilineH × 1), 7.16–7.00 (m, 4H, IndH5 + IndH2a + IndH2b + ^1^anilineH × 2), 6.96 (d, *J* = 8.8 Hz, 1H, IndH6), 6.77 (d, *J* = 8.1 Hz, 2H, ^1^anilineH × 2), 4.60 (d, *J* = 7.7 Hz, 2H, BnCH_2_), 3.53–3.43 (m, 2H + 2H + 2H, ^1^anilineCH_2_ × 2 + anilineCH_2_ × 2 + NCH_2_ × 2), 2.93–2.78 (m, 2H, IndCH_2_). MS (ESI) m/z [M + H] ^+^ calcd: 517.2; found: 517.2.^1^H NMR (400 MHz, DMSO-*d*_6_) δ 10.84 (s, 1H, IndH1), 8.15 (t, *J =* 4Hz, 1H, CONH), 7.54 (d, *J* = 7.9 Hz, 1H, IndH4), 7.33 (d, *J* = 8.1 Hz, 1H, IndH7), 7.17 (d, *J* = 2.3 Hz, 1H, IndH2), 7.05 (t, *J* = 7.4 Hz, 1H, IndH6), 6.95 (t, *J* = 7.4 Hz, 1H, IndH5), 6.88 (d, *J* = 7.9 Hz, 2H, Ar3,5), 6.47 (d, *J* = 8.1 Hz, 2H, Ar2,6), 4.94 (s, 2H, anilineNH_2_), 4.13–3.98 (dd, *J* = 8.0, 5.7 Hz, 2H, anilineCH_2_), 3.52 (s, 2H, IndCH_2_). MS (ESI) m/z [M + H] ^+^ calcd: 280.1; found: 280.0.^1^H NMR (400 MHz, DMSO-*d*_6_) δ 10.89 (s, 1H, IndH1), 9.40(s, 0.42H, PheOHa), 9.31(s, 0.58H, PheOHb), 7.53 (d, *J* = 7.7 Hz, 1H, IndH4), 7.40–7.30 (m, 1H, IndH7), 7.23–7.12 (m, 1H, IndH2), 7.12–7.02 (m, 1H, IndH6), 7.01–6.91 (m, 3H, IndH5 + anilineH3,5), 6.91–6.80 (m, 2H, PheH3,5), 6.78–6.54 (m, 2H + 2H, PheH2,4 + anilineH2,4), 4.43–4.24 (m, 2H + 2H, anilineNH_2_ + PheCH_2_), 3.83 (s, 2H, anilineCH_2_), 3.02 (ddd, *J* = 6.0, 2.9, 1.4 Hz, 1H, IndCH_2_ × 1), 2.82 (ddd, *J* = 6.0, 2.9, 1.4 Hz, 1H, IndCH_2_ × 1). MS (ESI) m/z M + H] ^+^ calcd: 386.2; found: 386.1.^1^H NMR (401 MHz, DMSO-*d*_6_) δ 10.89 (s, 1H, IndH1), 7.50 (dd, *J* = 21.2, 7.9 Hz, 1H, IndH4a + IndH4b), 7.44 (s, 1H, CONH_2_), 7.37–7.29 (m, 1.5H, IndH7 + IndH2a), 7.24 (s, 1H, CONH_2_), 7.15 (d, *J* = 7.6 Hz, 0.5H, IndH2b), 7.07 (t, *J* = 7.6 Hz, 1H, IndH6), 6.96 (t, *J* = 8.9 Hz, 1H, IndH5), 6.86–6.72 (m, 1H + 1H, ArH3,5a + ArH3,5b), 6.46 (m, 1H + 1H, ArH2,6a + ArH2,6b), 4.99 (d, *J* = 25.4 Hz, 2H, anilineNH_2_), 4.46 (d, *J* = 10.5 Hz, 1H, anilineCH_2_Ha), 4.31 (d, *J* = 10.7 Hz, 1H, anilineCH_2_Hb), 3.81 (d, *J* = 17.3 Hz, 2H, –NCH_2_), 3.70 (d, *J* = 21.7 Hz, 2H, IndCH_2_). MS (ESI) m/z [M + H] ^+^ calcd: 337.2; found: 337.0.^1^H NMR (401 MHz, DMSO-*d*_6_) δ 10.88 (s, 1H, IndH1), 7.56 (d, *J* = 7.9 Hz, 0.56H, IndH4a), 7.49 (d, *J* = 7.9 Hz, 0.44H, IndH4b), 7.34 (dd, *J* = 8.2, 2.7 Hz, 1H, IndH7a + IndH7b), 7.15 (dd, *J* = 8.8, 2.3 Hz, 1H, IndH2a + IndH2b), 7.07 (t, *J* = 7.5 Hz, 1H, IndH6), 6.96 (t, *J* = 7.4, 1H, IndH5), 6.90–6.78 (m, 2H, anilineH3,5), 6.66–6.34 (m, 2H, anilineH2,4), 4.38 (d, *J* = 30.3 Hz, 2H, anilineCH_2_), 3.78 (d, *J* = 15.3 Hz, 2H, IndCH_2_), 3.44–3.20 (m, 2H + 2H, αCH_2_ × 2 + γCH_2_ × 2), 1.68–1.57 (m, 1H, βCH_2_ × 1), 1.57–1.45 (m, 1H, βCH_2_ × 1). MS (ESI) m/z [M + H] ^+^ calcd: 338.2; found: 338.1.

#### 3.1.3. Radioligand Binding Assays

To determine the affinity of peptide analogs at the respective receptors, competition binding experiments were performed. Commercial membranes of CHO cells stably expressing human opioid receptors and the competing radioligands, [^3^H]DAMGO, [^3^H]deltorphin-2, and [^3^H]U-69593 for μ-OR, δ-OR, and κ-OR, respectively, were used. Membranes were incubated in 0.5 mL volume of 50 mM Tris/HCl (pH = 7.4), 0.5% bovine serum albumin (BSA), with a number of peptidase inhibitors (bacitracin, bestatin, captopril) and various concentrations of radioligands for 2 h at 25 °C. Non-specific binding was assessed in the presence of 10 mM naloxone. Three independent experiments for each assay were carried out in duplicate. The data were analyzed by a nonlinear least square regression analysis computer program Graph Pad PRISM 6.0 (Graph Pad Software Inc., San Diego, CA, USA). The IC_50_ values were determined from the logarithmic concentration-displacement curves, and the values of the inhibitory constants (K_i_) were calculated according to the equation of Cheng and Prusoff.

#### 3.1.4. Molecular Docking

The κ-OR model was extracted from the deposited X-ray structure human kappa opioid receptor in complex with JDTic, PDB ID 4DJH. To obtain the initial structure of each ligand, a geometry optimization was performed with the program ORCA 4.0 [41] using a Hartree–Fock method and a 6–31 G * basis set. Ligands and receptor were further processed using the AutoDock Tool Kit (ADT) [42] to obtain PDBQT format files.

The molecular docking simulations were performed with the AutoDock Vina 1.2.2 [43] software, with grid dimensions of 18 × 18 × 18 Å and grid position centred on the orthosteric binding site. The default values were used for the other parameters, except for the parameter exhaustiveness, set to 32. For each ligand, several docking runs were performed with each of the possible roots. To identify the most representative poses for each ligand, a cluster analysis was performed on all poses using ClusDOCK tool [44,45] from the PacDOCK web server. The gromos algorithm [46,47] was employed, with a cut-off of 2.0 Å and a minimum cluster size of 1 structure. The most representative poses of the most populated clusters associated with the most favorable scoring functions (kcal/mol) were considered.

### 3.2. Biology

#### 3.2.1. Cell culture and Treatments

HT29 human colorectal adenocarcinoma cells, HeLa human cervix adenocarcinoma cells, MCF7 human breast adenocarcinoma cells, PC-3 human prostate adenocarcinoma cells, J6 Jurkat Clone E6-1 acute T cell leukemia, the healthy I407 human intestine cells, and mouse embryonic fibroblast 3T3 fibroblasts as controls cell lines were purchased from American Type Culture Collection (ATCC, Manassas, VA). The human ovarian cancer cell line IGROV1 has been kindly provided by Prof. Colnaghi (Istituto Nazionale Tumori (IRCCS) (Milano, Italy). Cells were cultured in RPMI 1640 medium (Labtek Eurobio, Milan, Italy), supplemented with 10% FCS (Euroclone, Milano, Italy) and 2 mM L-glutamine (Sigma-Aldrich, Milano, Italy), at 37 °C, and a 5% CO_2_ atmosphere. The compounds were dissolved in DMSO in a 30–40 mM stock solution. In cell treatments, the final DMSO concentration never exceeded 0.1%.

#### 3.2.2. MTT Assay

Cells were seeded at 1.5 × 10^4^ cells/well in a 96-well culture plastic plate (Sarsted, Milan, Italy), and after 24 h of growth were exposed to increasing concentrations of each distinct compound (from 0.010 μM to 500 μM) solubilized in RPMI 1640 medium. MTT assay was performed according to the literature [48]. In brief, after 24 h of treatment, the culture medium was replaced with 0.1 mL of 3-(4,5-dimethylthiazolyl-2)-2,5-diphenyltetrazolium bromide (MTT, Sigma-Aldrich, Milano, Italy) dissolved in PBS at the concentration of 0.2 mg/mL, and samples were incubated for 2 h at 37 °C. The absorbance at 570 nm was measured using a multiwell plate reader (Tecan, Männemorf, Switzerland), and data were analyzed by Prism GraphPad software, and expressed as IC_50_ μM.

#### 3.2.3. Cell Cycle Analysis

HT29 were plated at density of 20,000 cell/cm^2^ in dish and after 24 h treated with 1 or 5 μM of **5** for 24 h. The samples were prepared in according to Calonghi [49]. In brief, untreated and treated cells were detached, washed in PBS and the pellet was finally re-suspended in 0.01% Nonidet P-40 (Sigma-Aldrich, Milano, Italy), 10 μg/mL RNase (Sigma-Aldrich, Milano, Italy), 0.1% sodium citrate (Sigma-Aldrich, Milano, Italy), 50 μg/mL propidium iodide (PI) (Sigma-Aldrich, Milano, Italy), for 30 min at room temperature in the dark. Propidium iodide (PI) fluorescence was acquired on a linear scale and analysed by Modfit software version 5.2 (USA). Flow cytometric assays were performed on a Brite HS flow cytometer (Bio-Rad, Watford, UK) equipped with a Xe/Hg lamp.

#### 3.2.4. Histone Post-Translational Modification

HT29 cells were seeded in dish and after 24 h they were treated for 6 h with compound **5** at final concentration of 5 µM. Cells were harvested, washed with 10 mM sodium butyrate in PBS, and nuclei were isolated in according to Amellem and Micheletti [50,51]. The nuclear pellet was suspended in 0.1 mL ice-cold H_2_O using a Vortex mixer, and concentrated H_2_SO_4_ was added to the suspension to give a final concentration of 0.4 N. After incubation at 4 °C for 1 h, the suspension was centrifuged for 5 min at 14,000× *g*, and the supernatant was taken and mixed with 1 mL of acetone. After overnight incubation, the coagulate material was collected by microcentrifugation and air-dried. This acid soluble histone fraction was dissolved in 20 μL of H_2_O. Proteins were quantified using a protein assay kit (Bio-Rad, Hercules, CA, USA). Histones were detected resolving samples on a 10% gel in MES buffer at 200 V for 40 min. Western Blot was performed in transfer buffer at 100 V for 1 h. The nitrocellulose membrane was incubated with primary antibody specific for anti-acetylated lysines (Millipore, Billerica, MA, USA) for 1 h. After five washes with PBS-TWEEN 20 0.1%, the membrane was incubated as before with secondary HRP-conjugated antibody (GE Healthcare, Milan, Italy). After five washes with PBS-TWEEN 20 0.1%, antibody binding was detected by Amersham ECL Plus Western Blotting Detection System (GE Healthcare, Milan, Italy).

#### 3.2.5. Quantitative Real-Time PCR

24 h after seeding, HT29 cells have been treated with 5 µM **5** for 6 h. Total RNA has been isolated by RNeasy Mini kit (Qiagen, Hilden, Germany) according to the manufacturer’s protocol. 1 μg of RNA has been reverse-transcribed with RevertAid First Strand cDNA Synthesis Kit (Fermentas, Ontario, Canada) by using oligo(dT) primers. The cDNA has been analyzed by quantitative Real-Time PCR (qRT-PCR), by employing the LightCycler FastStart DNA Master SYBR Green I Kit and the LightCycler 2.0 instrument (Roche Diagnostics, Manheim, Germany). Gene expression has been quantified by ΔΔC_T_ method, by using *G3PDH* as the housekeeping gene. The following primers were used: 5′-ATTTGGTCGTATTGGGCGCC-3′ (forward) and 5′-ACGGTGCCATGGAATTTGCC-3′ (reverse) for *G3PDH* detection, 5′-TGCAGACCCGGGAGAAAGATGT-3′ (forward) and 5′-ATCGAAATTCCACTTGCGCT-3′ (reverse) for *p27* detection 5′-CTAAGAGTGCTGGGCATTTT-3′(forward) and 5′-TGAATTTCATAACCGCCTGTG-3′ (reverse) for *p21 WAF1* detection, 5′-GATGCGTCCACCAAGAAGC-3′ (forward) and 5′-CCGCCACAAAGATGGTCAC -3′ (reverse) for *Bax* detection.

#### 3.2.6. Total Protein Extraction and Western Blot

HT29 cells were seeded in 25 cm^2^ flasks at a density of 2 × 10^4^ cells/cm^2^, and after 24 h, they were treated with 5 μM of compound **5**. After 24 h, both treated and control cells were washed twice with PBS and lysed in RIPA buffer [50 mMTris/HCl, 150 mMNaCl, SDS (1% v/v), Triton X-100 (1% v/v), 1 mM EDTA, pH 7.6]. Cell lysates were centrifuged at 12,000× *g* for 20 min, and the protein concentration was determined by using the Bio-Rad protein assay method (Bio-Rad, Hercules, CA, USA). The proteins were detected resolving samples on a 12% gel in Tris-Glycine buffer at 200 V for 50 min. Western Blot was performed in transfer buffer at 100 V for 1 h. The nitrocellulose membrane was incubated with primary antibodies for rabbit anti-p21 (Millipore, Billerica, MA, USA) or rabbit anti-p27 (Millipore, Billerica, MA) or mouse anti-Bax (Millipore, Billerica, MA, USA) or mouse anti-α tubulin (Invitrogen, Walthman, MA, USA) for 1 h. Detection of immunoreactive bands was performed by using a rabbit or mouse HRP-conjugated secondary antibody (GE Healthcare, Milan, Italy), followed by Amersham ECL Plus Western Blotting Detection System (GE Healthcare, Milan, Italy). Densitometry analysis of immunoreactive bands was done by Fluor-S Max MultiImager (Bio-Rad, Hercules, CA, USA). Relative quantification of bands was performed by using an α-tubulin signal as control.

#### 3.2.7. Statistical Analysis

All experiments were performed in triplicate and repeated at least three times. Results were averaged and the standard deviation was calculated. To determine statistical significance, unpaired two-tailed Student’s t test was used between two different independent groups. A *p*-value below 0.05 was considered significant.

## 4. Conclusions

In this work, we discuss the design of a mini-library of new indole-aryl amide derivatives, aiming at obtaining atypical opioid receptor ligands. Among the members of the mini-library, some compounds, including **5**, showed comparatively higher affinity for κ-ORs than for µ- and δ-OR. Albeit, the K_i_ was in the micromolar range, the binding experiments showed a good selectivity of **5** over the other ORs. Thereafter, the cytotoxicity of the compounds was assayed in a panel of cancer and healthy cell lines. The indole derivatives showed diverse activities, and in particular, **5** was shown to produce an effect on HT29 cells by inhibiting their proliferation and promoting their apoptosis, while sparing healthy I407 human intestine cells.

This led us to hypothesize that the molecular mechanism at the basis of the inhibitory effect of **5** might be related to the κ-OR. The interaction with this receptor could activate a signaling pathway with consequent epigenetic and gene expression modifications that lead to the inhibition of proliferation. Intriguingly, molecular docking simulations predict that the ligands interact with the message-binding region of the receptor by means of the indole ring.

In perspective, **5** could be further investigated as a novel candidate for colon cancer therapy development. However, we are aware that more research is needed to better define the functional **5** activities and future investigations should also consider recent findings, which strongly suggest the loss of mitochondrial integrity as a critical factor for the induction of cell death.

## Data Availability

Not applicable.

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
