# Peer review of "Synthesis and Antiproliferative Activity against Cancer Cells of Indole-Aryl-Amide Derivatives"

_molecules, 2022, doi:10.3390/molecules28010265_

Round 1

Reviewer 1 Report

The work has some advantages, but from my point of view it requires modifications, which I have outlined in the comments below:

·           Additional studies are necessary - the authors should compare the cytotoxic activity of the compounds with that of an FDA-approved anticancer drug on the same cell lines

·           Selectivity indices for the compounds - compared to both normal cell lines - should be provided in the additional table

·           Line 169 and Table 2: IC50 = 1643 μM - how is this possible since the compounds were tested at concentrations of 0.01-500 μM as stated in the text on line 159

·           The Figures 1 and 3, Scheme 2, Table 1 and 2 should be inserted into the main text close to their first citation

·           The order of the Figures in the Supplementary material is inconsistent with the citation in the manuscript

·           Figures S1-7 are not cited in the manuscript

·           Incorrectly cited references, e.g. there is: [1,2,3,4]; should be: [1-4]

Reviewer 2 Report

Although the derivatives are not many, the difference of the activity of the each compound was found.  The author should consider the effect of the chemical feature of amide, aniline, phenol, and aliphatic hydroxy group on the activity.  The effect of electron donation effect should be considered.  The effect of electron withdrawing group should be examined.  After these considerations, this referee recommend this article to publication. 

Author Response

REVIEWER 2 comments:
Although the derivatives are not many, the difference of the activity of the each compound
was found. The author should consider the effect of the chemical feature of amide, aniline,
phenol, and aliphatic hydroxy group on the activity. The effect of electron donation effect
should be considered. The effect of electron withdrawing group should be
examined. After these considerations, this referee recommend this article to publication.
Answer: we thank the Reviewer for the interesting suggestion. To address this point, we
introduced a discussion about the effects of the diverse substituents on the affinity for
kappa-opioid receptor. On the other hand, the definition of any structure-activity
relationship on the cytotoxicity and other biochemical effects at the moment is beyond our
possibility. In particular, because the molecular target has been not clearly identified yet.
This statement has been introduced in the revised version of the manuscript.

Round 2

Reviewer 1 Report

The current version of the manuscript is suitable for publication.